# Medical Residents’ Behaviours toward Compulsory COVID-19 Vaccination in a Tertiary Hospital in Italy

**DOI:** 10.3390/ijerph192315985

**Published:** 2022-11-30

**Authors:** Flavia Beccia, Giovanni Aulino, Carlotta Amantea, Alberto Lontano, Gerardo Altamura, Eleonora Marziali, Maria Francesca Rossi, Domenico Pascucci, Paolo Emilio Santoro, Antonio Oliva, Giovanni Capelli, Bruno Federico, Gianfranco Damiani, Patrizia Laurenti

**Affiliations:** 1Section of Hygiene, University Department of Life Sciences and Public Health, Università Cattolica del Sacro Cuore, 00168 Roma, Italy; 2Section of Legal Medicine, Department of Health Surveillance and Bioethics, Fondazione Policlinico A. Gemelli IRCCS, Università Cattolica del Sacro Cuore, 00168 Roma, Italy; 3Section of Occupational Health, University Department of Life Sciences and Public Health, Università Cattolica del Sacro Cuore, 00168 Roma, Italy; 4Department of Woman and Child Health and Public Health—Public Health Area, Fondazione Policlinico Universitario A. Gemelli IRCCS, 00168 Roma, Italy; 5National Center for Disease Prevention and Health Promotion, Italian National Institute of Health, 00161 Roma, Italy; 6Department of Human Sciences, Society and Health, University of Cassino and Southern Lazio, 03043 Cassino, Italy

**Keywords:** COVID-19, compulsory vaccination, residency, healthcare workers, Italian legislation

## Abstract

To maintain safety conditions in the provision of care and assistance, and to protect healthcare workers (HCWs) and patients, the Italian government required compulsory COVID-19 vaccination for HCWs, including medical residents (MRs). The aim of this study was to assess COVID-19 vaccination coverage in MRs in a large tertiary hospital in Italy, before and after the introduction of compulsory vaccination, according to demographic characteristics and specific residency. A database on COVID-19 vaccination status and infection of resident medical doctors was created. Descriptive statistics and logistic regressions were carried out on the data. A total of 1894 MRs were included in the study. Being vaccinated in the same hospital as the residency program was significantly related to the year of residency and being enrolled in a frontline residency. A significant association between compliance with the compulsory primary cycle vaccination and vaccination in the hospital residency was observed. Being enrolled in the second, third, and last years of residency, and in a frontline residency, were predictive of being vaccinated in the residency hospital. Almost 100% of the MRs participating in the study were vaccinated against COVID-19. Compulsory vaccination of HCWs, alongside greater and clearer information about the risks and benefits of vaccination, represents an important booster to ensure public health and to promote quality and safety of care.

## 1. Introduction

COVID-19 has shaped the world as we know it, presenting public health with unprecedented challenges, trying to contain the burden of disease and leveraging health protection interventions, both pharmacological and not, with political, social, and economic aspects on a global level [1]. Since the first detected case in Italy to date, almost 24 million cases have been reported, making Italy at the same time one of the first and most affected countries [2,3,4]. To address the pandemic, the Italian government enacted policies to slow the spread of the virus, offer economic support, and strengthen the health care system [5]. The advent of vaccination, about a year into the pandemic, has been the best driver to bring down COVID-19 mortality. Furthermore, Watson et al. estimated that vaccines prevented 19.8 million deaths in the first year of vaccination introduction with a 63% reduction in deaths caused by COVID-19 [6]. To maintain adequate safety conditions in the provision of care and assistance, while at the same time protecting healthcare workers (HCWs) and patients, the Italian government required compulsory vaccination for HCWs, both for the primary cycle, consisting of the first dose and second dose, and for the booster dose, through Decree Law 44/2021 and Decree Law 172/2021, respectively [7,8,9]. In case of non-compliance, in fact, the operator risks suspension from the practice of the profession, with the possibility of being assigned to another task that does not pose a risk of spreading contagion. In Europe, the imposition of compulsory vaccination has been particularly controversial, with the only examples in Italy and France [10,11]. Although the COVID-19 vaccine is unlikely to lead to the achievement of herd immunity due to both the rapidity of virus mutation and the short duration of the immune response, it has been shown to be effective in reducing the SARS-CoV-2 incidence and mortality, especially among the most fragile individuals [12,13,14]. Furthermore, vaccination contributes to the reduction in HCWs’ absences from work, ensuring the protection of HCWs themselves and of patients by guaranteeing the continuity of care services, also ensuring economic savings [15,16,17,18,19].

The COVID-19 pandemic and the resulting health crisis exacerbated the shortage of healthcare personnel employed in the National Health Service [20]. In order to deal with the emergency, it was necessary to implement extraordinary measures to recruit personnel. These measures aimed at strengthening territorial care networks and hospital departments for the care of COVID-19 patients and the implementation of the COVID-19 vaccination campaign [21]. During the early stages of the pandemic, the Decree Law 9/2020 allowed, on a transitional basis and as an exception to existing regulations, the conclusion of self-employment contracts with healthcare personnel in highly critical territories [22,23]. Subsequently, as the epidemiological situation worsened, Article 2-bis of the Decree Law 18/2020 extended the possibility of hiring physicians, nurses, and other healthcare professionals on self-employment contracts to the entire country [24]. The legislation has been extended several times, most recently to 31 December 2022 by Law 234/2021 [25]. Medical personnel also included medical residents, who are enrolled in the last and penultimate years of the specialisation schools. According to the Court of Auditors’ Public Finance Report 2021, 4068 freelance contracts were signed with medical residents in 2020 [26]. They have therefore played a crucial role in the fight against the pandemic, often finding themselves on the front lines dealing with SARS-CoV-2.The aim of this study is to investigate SARS-CoV-2 vaccination coverage among the medical residents (MRs) of a large tertiary hospital in Italy, and to evaluate whether age, gender, and type of residency influenced the timing of vaccination for both primary cycle and booster dose, in relation to the legal obligation and the campaign conducted in the same hospital of residency.

## 2. Materials and Methods

To serve the study’s objectives, a database on COVID-19 vaccination status and infection of medical doctors enrolled in a residency program at Fondazione Policlinico A. Gemelli IRCCS (FPG) was created. Vaccination data, collected routinely during the COVID-19 vaccination campaign in FPG, were used. They consisted of dates of first and second vaccination (hereby referred to by “primary cycle”) and booster vaccination date. The only vaccine administered was the vaccine produced by Pfizer-BioNTech “Comirnaty”, with the vaccine schedule consisting of two doses per primary cycle and a single dose as a booster; all doses were administered free of charge for the MRs. Available data regarding swabs conducted within the facility, through the surveillance system provided for the facility’s HCWs, were associated with the MRs’ vaccination data at the time of database creation, using the matriculation numbers as linkage. The data were pseudonymised and processed in compliance with the consents provided and the relevant data processing regulations. Dispositions in DL n.44/2021 (Law 76) and DL 172/2021 were used to categorise data on vaccination date as completed before or after 1st April 2021, when the compulsory policy became effective for the primary cycle, and before and after 15 December 2021, when the policy for the booster dose became effective.

To avoid classification bias, data on infection were used to adjust the statistical analysis, because people with infection since 1 January were excluded from the analysis on the primary cycle and people with infection up to 15 August were excluded from the analysis on the booster dose, in line with ministerial decree, as they could not have been administered a vaccination dose so close to the COVID-19 infection, even if they intended to be vaccinated in compliance with Law 76. These data were collected from the surveillance system available in the hospital for screening healthcare personnel according to regional and national dispositions. All the results are presented as descriptive data.

For each participant, the residency field variable, reporting the specific residency in which each doctor is enrolled, was coded into frontline and not-frontline (Appendix A) in managing COVID-19 patients or directly involved in the vaccination campaign. Furthermore, considering that some residencies comprise four years, while others are five years, the fourth- and fifth-year were aggregated for inferential statistics.

Descriptive analyses were performed for all sociodemographic variables (gender, age, year of residency, and vaccination in or outside the residency hospital). Univariate and logistic regressions for each of the outcomes considered were carried out, with a statistical significance of *p* < 0.05. All the analyses were performed with STATA 14 software.

The study protocol was approved by the FPG Ethical Committee (ID3973).

## 3. Results

In total, 1894 MRs were included in the study. The sample was comprised of 854 (45.09%) male and 1040 (54.91%) female MRs, and the mean age was 29.20 (SD ± 3.27) years; most of the participants (54.07%) were in their first year of residency, were not frontline (70.43%) HCWs, and were vaccinated in the same hospital where they were completing their residency (57.28%). Of the 1889 MRs who were vaccinated for the primary cycle (99.73%) and 1867 for booster dose vaccination (98.83%), 56% (1067 and 1054 MRs, respectively) did so in the residency hospital. Adjusting by the decree regulating the compulsory vaccination, 1870 MRs for the primary cycle and 1842 MRs for the booster dose were considered in the statistical analysis (Table 1).

Five MRs were not vaccinated against COVID-19. Their mean age was 33.8 years (SD 8.84), and they were all in their first year of residency; two were males (40.00%) and three were females (60.00%). Two (40.00%) of them were frontline MRs (general surgery and emergency medicine), while three (60.00%) were not frontline MRs (occupational health, internal medicine, and neurology).

Being vaccinated in the same hospital as the residency program was significantly related to year of residency (*p* < 0.001) and being enrolled in a frontline residency (*p* < 0.001), but not to gender (Table 2). Our results showed that 80.08% of third-year MRs were vaccinated in their hospital, but the same was true only for 44.85% of first-year MRs. Furthermore, 63.62% of frontline specialties’ MRs were vaccinated in the hospital they were attending, while the same was true for 54.62% of non-frontline specialties’ MRs.

Our results showed a statistically significant association between compliance with the compulsory primary cycle vaccination and being vaccinated in the same hospital in which the resident was working (*p* < 0.001), highlighting that only 4.78% of MRs vaccinated in their own hospital received their vaccine after the compulsory policy became effective, while 12.95% of MRs vaccinated elsewhere received their vaccination after April 1st, therefore not complying with the compulsory vaccination (Table 3).

Concerning booster dose vaccination, a statistically significant association was also found with being vaccinated in the same hospital as the residency (*p* = 0.007), with 9.68% of MRs being vaccinated in their own hospital not complying with the compulsory booster vaccination policy, against 13.71% of MRs being vaccinated elsewhere (Table 3).

Concerning the logistic regression models, being enrolled in the second, third, and last years of residency (*p* < 0.001) and being enrolled in a frontline residency [OR 1.65, CI (1.33–2.04), *p* < 0.001] were predictive of being vaccinated in the residency hospital (Table 4).

Being vaccinated in the residency hospital [OR 0.38, CI (0.27–0.55), *p* < 0.001], and being enrolled in the third (*p* = 0.006) or last year (*p* = 0.039) of residency were predictors of becoming vaccinated before the compulsory vaccination policy for the primary cycle became effective (Table 4).

Finally, being vaccinated after the compulsory policy for primary cycle [OR 12.90, CI (8.66–19.23), *p* < 0.001] was a predictor of receiving the booster vaccination dose after the compulsory policy for it became effective; being a frontline resident was associated with becoming vaccinated before the compulsory policy became effective [OR 0.63, CI (0.44–0.92), *p* = 0.016] (Table 4).

## 4. Discussion

According to our findings, the rate of resident doctors who adhered to the COVID-19 vaccination programme is close to 100%; in fact, only 0.26% of them did not receive a vaccination. These results show a very high compliance rate of MRs towards compulsory vaccination even compared to studies performed in other states [27,28,29,30,31].

Vaccination coverage expected for compulsory vaccinations in HCWs is close to 90%, whereas it is about 40% for other recommended vaccinations [32,33]. Therefore, our results are above the expected when taking into consideration the population and the fact that COVID-19 vaccination was compulsory in Italy at the time of the study. This result could be related to the fact that younger doctors are more likely to adhere to vaccination campaigns. However, a comparison to adherence rate for other vaccinations—despite the same population (i.e., HCWs) being studied—is not entirely possible, due to the social and public health importance that the COVID-19 vaccination currently has. To expand on this, the compulsory vaccination in Italy has been enforced (by suspending non-vaccinated HCWs from duty) up until recently when the compulsory vaccine programme was amended. Due to this, and due to the importance placed on vaccination while the pandemic is still ongoing, a comparison to other vaccination campaigns falls short, although the nearly 100% participation rate in this study showcases the importance that vaccination has for younger doctors as a public health measure [34].

Furthermore, in our study we investigated the vaccination rate before and after the compulsory vaccination imposed by Decree Law 44/21 and 172/21: more than 90% and about 88% of the MRs had been vaccinated before the introduction of the compulsory vaccination for the primary cycle, and booster vaccination, respectively. However, it should be specified that being vaccinated after the compulsory policy for the primary cycle was a predictor of receiving the booster vaccination dose after the compulsory policy for it became effective. This might suggest a hesitancy to vaccinate on the part of this section of HCWs, which could be due on the one hand to a concern about the cost and safety of vaccines and on the other hand to a lack of adequate information from the government regarding vaccination policies [35,36].

Compulsory vaccination against COVID-19 has been particularly controversial, especially for vaccine deniers, but also because of resolution no. 2361 (“COVID-19 vaccines: ethical, legal and practical considerations”) passed by the Council of Europe, the main international organisation committed to protecting human rights, separate and independent from the European Union. It requires member states to “ensure that citizens are informed that the vaccination is not compulsory and that no one is under political, social or other pressure to be vaccinated if they do not wish to do so; ensure that no one is discriminated against for not having been vaccinated, due to possible health risks or not wanting to be vaccinated” [37]. It must be made clear that it is not a source of law and that no member state is required to abide by it. Moreover, it is stated that, in the same way as other nations, Italy had a clear need to impose vaccination requirements for HCWs as a response to a pressing societal need for the protection of individual and public health and, more importantly, as a defence for weaker individuals or patients, for whom HCWs, such as resident doctors, have a specific position of guarantee and trust [38,39]. For these reasons, we believe that the vaccination requirement for HCWs was a necessary boost to ensure public health protection.

Alongside compulsory vaccination, a necessary boost to vaccination adherence is represented by the awareness and vaccination campaigns [17]. In our case, almost 60% of the MRs were vaccinated in the hospital of their specialisation thanks to the organisation of these campaigns; among MRs, the “frontline” are the ones who have used this service the most in the hospital where they work.

In fact, our study compared adherence to vaccination of both frontline and non-frontline workers, demonstrating a statistically relevant higher number of MRs receiving the shot inside the residency hospital for the first group, coherently with the policy to prioritise the most exposed workers. Comparing different models demonstrated that vaccinating essential workers sooner had strong benefits in terms of reducing infections, hospitalisations, and deaths, and in terms of net monetary benefit [40].

Furthermore, frontline specialisations and younger doctors (first-year) were among those who vaccinated first for the booster dose. However, it is important to emphasise that the opposite happened for the primary dose: in fact, non-frontline specialisations and older doctors vaccinated first.

This result may have been due to several factors: on the one hand, it may have been due to the greater involvement of the MRs in the awareness campaigns organised for them; on the other hand, it may have been due to the fact that a higher proportion of infections among frontline MRs caused the start of their vaccination cycle to be delayed.

These results could provide some insight into the behaviours of young doctors, and the influence of some factors on their vaccination coverage. It would be very interesting to explore, in future studies, the association between COVID-19 vaccination and other vaccinations, to explore the knowledge of and attitudes to vaccinations, and to further analyse the influence of policies and laws on preventive measures and public health. Furthermore, it is important to consider that the compulsion on COVID-19 vaccination has expired in Italy starting 1st November 2022 (Legislative Decree 162, 31 October 2022). This could have consequences on the uptake of the second booster dose (fourth dose of the COVID-19 vaccination), since HCWs in general may be less inclined to adhere to the vaccination campaign. Despite this, results from our study highlight that the vaccination uptake in MRs largely happened before the vaccination was made compulsory, therefore the uptake could remain significant for future vaccination. For these reasons, a follow-up evaluation of vaccination uptake for the second booster dose of the vaccine—and possible future additional doses—would be very interesting; such an assessment will further distinguish between the merit of compulsory vaccination and the intent to vaccinate in MRs due to other factors (such as the knowledge about vaccination benefits), which seems to have played an important factor in the current study since the vaccination uptake was high even before the vaccination was compulsory.

This study should be considered in light of some limitations and strengths. It is possible that the data on COVID-19 tests, on which vaccine uptake latency was calculated, were not complete, as only data on swabs performed or reported to the facility of affiliation (FPG) were collected. In addition, it is possible that a proportion of MRs may have been vaccinated outside the referring hospital due to a shortage of available slots, since booking to receive a vaccination followed an in-house online booking system. Furthermore, first-year MRs, entering residency during the course of the vaccination campaign, may have preferred to vaccinate at another centre, in line with what they did for the primary cycle. “No-vax” MRs may have been vaccinated in another region, so a lack of communication between regional information systems may have affected the availability of information. The present study does not evaluate the efficacy of vaccination campaigns and policies, but it investigates MRs’ behaviours, without exploring causal relationships. This study tries to provide an indicator of behaviour pattern distribution and an analysis of some factors that could influence the distribution. In addition, the lack of an external comparator and the unstudied reasons for the behaviours make it difficult to accurately assess the impact and effectiveness of compulsory vaccination policies on MRs. Nevertheless, this study represents the first example of an analysis of MRs’ behaviours toward compulsory vaccination, both primary cycle and booster dose, valuing residency and demographic characteristics. Building on the results presented, vaccine strategies could be adjusted by residency, increasing effectiveness.

## 5. Conclusions

Italy has been one of the countries hardest hit by the pandemic, and the healthcare system’s ability to cope with it has been put to the test, firstly with patient management and subsequently with the organisation and management of vaccination campaigns. The compulsory vaccination has been important to ensure the rapid and large-scale adhesion to the COVID-19 vaccination campaign by HCWs, including MRs. In association, vaccination campaigns in hospitals, associated with awareness campaigns about the risks and benefits of the vaccination, have been instrumental in achieving adequate vaccination coverage in the population considered. For these reasons, we believe that only the synergy of these two policies, together with greater and clearer information on the risks and benefits of vaccination, represents an important driving force for guaranteeing public health during the period of a health emergency.

The residency hospitals need to care for their MRs, always offering them vaccinations, by considering their engagement in care and their value as young leaders and linked professionals.

## Figures and Tables

**Table 1 ijerph-19-15985-t001:** Sociodemographic characteristics of the sample.

		Mean	Standard Deviation
**Age**		29.20	3.27
		**Number**	**Percentage**
**Gender**	**Male**	854	45.09%
	**Female**	1040	54.91%
**Year of residency**	**1**	1024	54.07%
	**2**	301	15.89%
	**3**	241	12.72%
	**4 or 5**	328	17.32%
**Frontline residency**	**No**	1334	70.43%
	**Yes**	560	29.57%
**Vaccination in residency hospital**	**No**	807	42.72%
	**Yes**	1082	57.28%

**Table 2 ijerph-19-15985-t002:** Univariate analyses were performed using the Chi-Square test, with vaccination in residency hospital as the outcome and year of residency, frontline residency, and gender as covariables (* *p* < 0.05, statistically significant).

		% Vaccinated in Residency Hospital	*p*-Value
**Year of residency**	**1**	44.85%	<0.001 *
	**2**	75.42%	
	**3**	80.08%	
	**4 or 5**	62.50%	
**Frontline residency**	**No**	54.62%	<0.001 *
	**Yes**	63.62%	
**Gender**	**Male**	56.34%	0.454
	**Female**	58.05%	
	**Total**	57.28%	

**Table 3 ijerph-19-15985-t003:** Univariate analyses were performed using the Chi-Square test, with vaccination completed before compulsory policy (for primary cycle vaccination and booster dose), and vaccination in residency hospital as the covariate (* *p* < 0.05, statistically significant).

		Primary Cycle Vaccination after 1 April	*p* Value	Booster Dose Vaccination after 15 December	*p* Value
**Vaccination in residency hospital**	**No**	12.95%	<0.001 *	13.71%	0.007 *
	**Yes**	4.78%		9.68%	
	**Total**	8.29%		11.40%	

**Table 4 ijerph-19-15985-t004:** Logistic regression models for vaccination in residency hospital, primary vaccination before compulsory policy, and booster dose before compulsory policy (* *p* < 0.05, statistically significant).

		Vaccination in Residency Hospital		
		Odds Ratio	95% Confidence Interval	*p* Value
**Year of residency**	**1**	1		
	**2**	3.96	2.96–5.31	<0.001 *
	**3**	5.21	3.70–7.34	<0.001 *
	**4 or 5**	2.10	1.62–2.71	<0.001 *
**Frontline**	**No**	1		
	**Yes**	1.65	1.33–2.04	<0.001 *
**Gender**	**Male**	1		
	**Female**	1.09	0.90–1.32	0.368
		**Primary cycle vaccination after compulsory policy**		
		**Odds Ratio**	**95% Confidence Interval**	** *p* ** **value**
**Vaccination in residency hospital**	**No**	1		
	**Yes**	0.38	0.27–0.55	<0.001 *
**Year of residency**	**1**	1		
	**2**	0.53	0.32–0.89	0.233
	**3**	0.21	0.09–0.49	0.006 *
	**4 or 5**	0.50	0.30–0.82	0.039 *
**Frontline**	**No**	1		
	**Yes**	1.38	0.97–1.97	0.074
**Gender**	**Male**	1		
	**Female**	1.32	0.94–1.85	0.098
		**Booster dose vaccination after compulsory policy**		
		**Odds Ratio**	**95% Confidence Interval**	** *p* ** **value**
**Primary vaccination before compulsory policy**	**Before**	1		
	**After**	12.90	8.66–19.23	<0.001 *
**Vaccination in residency hospital**	**No**	1		
	**Yes**	0.87	0.62–1.22	0.412
**Year of residency**	**1**	1		
	**2**	1.23	0.78–1.93	0.378
	**3**	1.21	0.72–2.03	0.476
	**4 or 5**	1.10	0.70–1.73	0.670
**Frontline**	**No**	1		
	**Yes**	0.63	0.44–0.92	0.016 *
**Gender**	**Male**	1		
	**Female**	1.16	0.84–1.59	0.372

## Data Availability

Data are available upon reasonable request.

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
