# Peer review of "Medical Residents’ Behaviours toward Compulsory COVID-19 Vaccination in a Tertiary Hospital in Italy"

_ijerph, 2022, doi:10.3390/ijerph192315985_

Round 1
Reviewer 1 Report
The electronically submitted MS is an original article aiming to assess COVID-19 vaccination coverage in medical residents in a large tertiary hospital in Italy, before and after the introduction of compulsory vaccination, according to demographic characteristics and specific residency.
It is an, overall, well-written manuscript. The methodology is simple. The justification of the study and the methodology could be more developed. There could be more discussion on the potential implications and the significance of these results.
Below, I leave some comments, so that the authors could take them into consideration for a revised version of the MS:
Line 42; COVID-19 has shaped the world as we know it, presenting public health with an unprecedented challenge: please, be more specific.
Line 43; With 22.5 million cases, Italy is one of the most affected countries in Europe: please, clarify if it refers to the whole pandemic period, as Italy was amongst the first countries to be affected.
Line 50; HCWs: please, explain the abbreviation.
Introduction: there is no justification for the purpose and methodology of the study. Why is this study important? What do we pretend to learn from this? Why the results (for example, Table 2) of this study are important? Are there any other similar studies in the region and other parts of the world? And, why the authors aimed to study vaccination coverage by focusing exclusively on medical residents? Are there any data in the literature regarding vaccination against other viruses?
Lines 69-70; Among health care professionals, MRs deserve particular attention because they are 69 the professionals of the present and of the future in which we must invest: Health care professionals are a group that does not include only medical doctors. Also, the professionals of the present are many more than the study group. I believe that the authors should consider revising this phrase. Also, it could be better if the last part of the introduction is the objective. Any justification for the study should be mentioned before the objective.
Line 82: specify what type of bias.
Line 96: specify types of the variable “place of vaccination”.
Lines 101-105: It could be interesting to indicate the overall % of vaccination (first and booster).
Overall (Results, Methodology, and Discussion): According to the authors, five MRs were not vaccinated against COVID-19 (the rate of resident doctors who adhered to the COVID-19 vaccination is close to 100%, in fact, only 0.26% of them did not vaccinate). So, is it possible that statistical differences can be attributed to other factors? Did the authors adjust for some variables, as the population study is not homogenous (as Table 1 indicates)? What exactly do your results show? And, if any statistical differences are significant, what are the clinical (or other) implications of these findings? The following limitation (already mentioned in the discussion) should also be considered when interpreting the results of these analyses: “No-vax” MRs may have been vaccinated in another region, so a lack of communication between regional information systems may have affected the availability of information.
Lines 173-175: does the methodological approach of this study support this statement? Please, revise this phrase to avoid potentially misleading assumptions.
Conclusions: does the methodological approach of your study support these conclusions?
Author Response
Dear reviewer,
Thank you for your valuable advice. After reading your comments we edited our work in the following points. A detailed answer to all the comments and the changes made are reported below:
Comment: “Line 42; COVID-19 has shaped the world as we know it, presenting public health with an unprecedented challenge: please, be more specific.”
Response: According to the suggestion, we expanded on the impact of COVID-19 pandemic on public health (lines 43-48); in addition, a new reference was added [1].
Comment: “Line 43; With 22.5 million cases, Italy is one of the most affected countries in Europe: please, clarify if it refers to the whole pandemic period, as Italy was amongst the first countries to be affected.”
Response:
Thank you for your input, we clarified the meaning of this sentence (lines 48-52), adding the Italian COVID-19 data repositories as reference [2].
Comment: “Line 50; HCWs: please, explain the abbreviation.”
Response: The abbreviation was explained.
Comment: “Introduction: there is no justification for the purpose and methodology of the study. Why is this study important? What do we pretend to learn from this? Why the results (for example, Table 2) of this study are important? Are there any other similar studies in the region and other parts of the world? And, why the authors aimed to study vaccination coverage by focusing exclusively on medical residents? Are there any data in the literature regarding vaccination against other viruses?”
Response: Thank you for your valuable comment. The aim of the study was to assess COVID-19 vaccination coverage among medical residents (MRs), and to evaluate whether age, gender and type of residency influenced the timing of vaccination in relation to the legal obligation and the campaign conducted in the same hospital of residency. This study is the first of its kind, so comparing data obtained to other MRs samples is not possible. However, we have attempted to make a comparison with data reported generically on healthcare workers (which we know well does not only include MRs), making two main assumptions. The first is that the MRs in the hospital considered are an adequate representative sample, considering that they number 1894 out of approximately 7000 employed healthcare workers. The second is that, considering the distribution of age, gender, and type of residency, the sample could also be representative of the Italian MRs, who were extensively involved in the management of the pandemic, both on the organisational and clinical side. This is the main reason why we have chosen this specific population, and why we believe that it is of extreme interest to the medical profession in Italy to organise vaccination campaigns and investigate what factors may influence vaccination adherence. Of course, our analysis has some limitations: the lack of data on other healthcare providers and the aforementioned lack of comparison of the results obtained with literature data on similar samples. By presenting these results, we hope to provide some insight into the behaviours of young doctors, and the influence of some factors (some related to the residency, some to the pandemic) on their vaccination coverage. It would be very interesting to explore, in future studies, the association between COVID-19 vaccination and other vaccinations, like hepatitis, flu, measles, rubella and chickenpox, and other vaccine-preventable diseases. this type of study, although already in the authors' intentions, turned out not to be feasible, as these data are collected by the university institution and not by the hospital (thus foreseeing a different process for approval by the ethics committee) and the consent administered was only related to the COVID-19 vaccination. We would like to stress that the aim of the study is not to estimate or evaluate the vaccination coverage per se, but the coverage and behaviour of the MRs.
Building on your suggestion, we expanded these considerations in the introduction (lines 73-89) and discussion (lines 215-228) section.
Comment: “Lines 69-70; Among health care professionals, MRs deserve particular attention because they are 69 the professionals of the present and of the future in which we must invest: Health care professionals are a group that does not include only medical doctors. Also, the professionals of the present are many more than the study group. I believe that the authors should consider revising this phrase. Also, it could be better if the last part of the introduction is the objective. Any justification for the study should be mentioned before the objective.”
Response: According to the reviewer’s indication, we rephrased the objective of the study (lines 90-99).
Comment: “Line 82: specify what type of bias.”
Response: Thank you for your comment: by adjusting the identification of different groups using the available data on infections, we tried to avoid classification bias, and we have added it in the manuscript (line 117).
Comment: “Line 96: specify types of the variable “place of vaccination”.”
Response: We specified we were referring to the residency hospital (line 131).
Comment: “Lines 101-105: It could be interesting to indicate the overall % of vaccination (first and booster).”
Response: According to reviewer’s comment we indicated the overall % of vaccination (linea 142-146).
Comment: “Overall (Results, Methodology, and Discussion): According to the authors, five MRs were not vaccinated against COVID-19 (the rate of resident doctors who adhered to the COVID-19 vaccination is close to 100%, in fact, only 0.26% of them did not vaccinate). So, is it possible that statistical differences can be attributed to other factors? Did the authors adjust for some variables, as the population study is not homogenous (as Table 1 indicates)? What exactly do your results show? And, if any statistical differences are significant, what are the clinical (or other) implications of these findings? The following limitation (already mentioned in the discussion) should also be considered when interpreting the results of these analyses: “No-vax” MRs may have been vaccinated in another region, so a lack of communication between regional information systems may have affected the availability of information.”
Response: Thank you for the valuable comment. We do not evaluate the effectiveness of the campaign or policies, only the behaviors, without delving into the causal relationship. This study provides an indicator of distribution of behaviors and an analysis on some factors that might influence the distribution. We did not find, in statistical analysis, significant differences between the groups created for the demographic characteristics considered, and we assessed the presence of confounding factors. In addition, we considered data on infections precisely to avoid misclassification. The results show that young people in frontline (i.e., afferent to medical specialties most in contact with COVID-19 patients or directly involved in the organization of the vaccination campaign) are more likely to be vaccinated at the place of employment. The results are objective and obtained from the data in hand, according to the methodology presented; instead, we presented the key to interpret the results in the discussion (lines 277-294), recognizing the limitations in the methodology adopted (lines 304-310). The clinical and nonclinical implications of the results obtained are that vaccination policies and campaigns (thus macro- and meso-level actions) can be extremely functional in reducing the burden of vaccine-preventable diseases.
Comment: “Lines 173-175: does the methodological approach of this study support this statement? Please, revise this phrase to avoid potentially misleading assumptions.”
Response: Thank you for noting the lack of clarity in the sentence and the inconsistency with the methodology used. We have removed it and specified this aspect within the limits of the study.
Comment: “Conclusions: does the methodological approach of your study support these conclusions?”
Response: Thank you very much for pointing that out. We revised the conclusion accordingly (lines 316-326).
Reviewer 2 Report
Dear authors of manuscript ijerph-2020273: Attitudes toward compulsory COVID-19 vaccination of medical residents in a tertiary hospital in Italy.
This manuscript is presenting a study of descriptive design dealing with vaccination of Health Care Workers (HCW) with covid-19 vaccine, at a tertiary hospital before and after decision on compulsory HCW-vaccination in Italy. Given the fact that compulsory vaccination can be questioned as an effective tool for vaccine trust and future vaccination campaigns it is interesting to study the effect in HCW
My main comments are:
1. No data is presented regarding the information collected on covid-19 infection from the hospital screeing system, as far as I can see? As previous infection influenced the need to follow the compulsory vaccination and also when that was done later this would have been interesting to see, especially as this is mentioned in the Methods, then the data should be presented.
2. Also, regarding hospital surveillance system on covid-19 testing for HCW how was this linked with the vaccination register, as it is stated that the latter was anonymized? Please explain in Methods.
3. Please provide background data on HCW-vaccination rate pre-covid-19 (influenza or measles?) or at other similar hospitals. Also, vaccination coverage in other HCW or similar (age/gender) population groups in the society would be useful to compare. It is otherwise difficult to put this in context for the reader. See also comment below about external validity.
4. Please provide information about experienced adverse events after primary vaccinations and if linked to age, geneder, previous infection or ability to take booster dose.
5. Please provide clear information if the cost of the vaccination was fully reimbursed and which vaccine was used.
6. Please in conclusion separate what is found in this study and speculations/feelings and if, or if not , any practical conclusion can be drawn from this given the fact that most medical residents were already vaccinated before compulsory vaccination. Did compulsory vaccination increase the ability to prevent the disease by HCW to fragile patients?
7. External validity: How valid are your data outside this hospital and/or outside Italy?
Author Response
Dear reviewer,
Thank you for your valuable advice. After reading your comments we edited our work in the following points. A detailed answer to all the comments and the changes made are reported below:
Comment: “No data is presented regarding the information collected on covid-19 infection from the hospital screeing system, as far as I can see? As previous infection influenced the need to follow the compulsory vaccination and also when that was done later this would have been interesting to see, especially as this is mentioned in the Methods, then the data should be presented.”
Response: Infection data were collected for the sole purpose of avoiding misclassification of pre- and postobligation groups (methods section, lines 117-124). Looking at the Results, the infection data led to the exclusion of 19 MRs for the primary cycle and 25 MRs for the booster dose from the analyses. It was only a precaution stipulated in the study protocol, but it did not significantly influence the results. Unfortunately, we did not plan to present data on infections per se, as it was not instrumental to the objectives of the study.
Comment: “Also, regarding hospital surveillance system on covid-19 testing for HCW how was this linked with the vaccination register, as it is stated that the latter was anonymized? Please explain in Methods”
Response: Thank you for pointing out the lack of clarity of the process. Data on infection and vaccination were linked with the matriculation numbers. We expanded on this aspect in the Methods (lines 108-112).
Comment: “Please provide background data on HCW-vaccination rate pre-covid-19 (influenza or measles?) or at other similar hospitals. Also, vaccination coverage in other HCW or similar (age/gender) population groups in the society would be useful to compare. It is otherwise difficult to put this in context for the reader. See also comment below about external validity.”
Response: Thank you for your valuable suggestions, we expanded these aspects in the discussion (lines 215-228).
Comment: “Please provide information about experienced adverse events after primary vaccinations and if linked to age, geneder, previous infection or ability to take booster dose.”
Response: Thank you for your valuable input. In this study, we did not analyze the adverse effects of vaccination, nor would it have been possible without follow-up survey integrated in the pharmacovigilance systems and processes. Adverse events are reported to the Italian Medicines Agency (AIFA), and by using the matriculation number instead of other data, it is not possible for us to obtain adverse events data from pharmacovigilance databases, considering that the reporting of adverse events is anonymous. In addition, the methodology adopted did not involve the administration of questionnaires or linkage with databases external to the foundation. We treasure your suggestion for future studies on vaccination and MRs.
Comment: “Please provide clear information if the cost of the vaccination was fully reimbursed and which vaccine was used.”
Response: Comirnaty vaccine was the only vaccine undertaken in the hospital, it was offered free of charge. We added more details on this aspect in methods section (lines 106-108).
Comment: “Please in conclusion separate what is found in this study and speculations/feelings and if, or if not , any practical conclusion can be drawn from this given the fact that most medical residents were already vaccinated before compulsory vaccination. Did compulsory vaccination increase the ability to prevent the disease by HCW to fragile patients?”
Response: Thank you very much for pointing out the lack of clarity in the conclusion. We revised the conclusion accordingly (lines 317-327).
Comment: “External validity: How valid are your data outside this hospital and/or outside Italy?”
Response: The study was not conducted on a sample of MRs, but on the entire population of MRs at the indicated hospital. Based on this assumption, we can draw the conclusion that the sample is representative of the Italian population of MRs in light of the following factors: 1) the referral hospital, by virtue of the volumes and activities of the services provided in the health care system, offers a panorama of training offerings that covers all possible branches that are the subject of medical specialization; 2) the ratio of enrollment in each residency school is consistent with data from other schools nationwide.
Reviewer 3 Report
Please clarify frontline workers definition in line 194-195. The outcome given in abstract and conclusion are not inline, need improvements. The summary of the results should be mentioned in abstract to make it more interesting. The risk and benefits of vaccine as described in abstract is not detailed in body of the article. Are you talking about campaign? Overall, it is a simple nice study.
Author Response
Dear reviewer,
Thank you for your valuable advice. After reading your comments we edited our work in the following points. A detailed answer to all the comments and the changes made are reported below:
Comment: “Please clarify frontline workers definition in line 194-195.”
Response: To make sure the main text of the manuscript is easier to understand, we have detailed the frontline and non-frontline MRs classification in the supplementary material Table 1, as we described in the methods section of the manuscript (lines 127-128).
Comment: “The outcome given in abstract and conclusion are not inline, need improvements.”
Response: We have rephrased the conclusions (lines 317-327) section to be more in line with the abstract, thank you.
Comment: “The summary of the results should be mentioned in abstract to make it more interesting.”
Response: We added a sentence summarizing our results to make the abstract more interesting (lines 36-37), thank you.
Comment: “The risk and benefits of vaccine as described in abstract is not detailed in body of the article. Are you talking about campaign?”
Response: In the abstract, we mention the importance of information about risks and benefits of vaccination (we did not state that we measured it in the study), meaning that a more detailed information campaign could be effective in encouraging more HCWs to get vaccinated – although of course not as effective as compulsory vaccination, which was implemented in Italy at the time of the study. To be clearer about the meaning of this sentence, we expanded on it in the conclusion section of the manuscript (lines 319-327), thank you for your valuable input.
Reviewer 4 Report
Beccia F et al. present a quite interesting paper regarding attitudes toward compulsory COVID-19 vaccination of Italian medical residents. It may be of interest for epidemiologists and public health experts. The manuscript is well structured and written. It is suitable for publication in the International Journal of Environmental Research and Public Health, however I would like to make few suggestions that might be considered in the final version. 1) It would be helpful to provide a bit more detailed information regarding how attitudes were assessed (questionnaires?) as raw data from database regarding the fact who, when and where underwent vaccination is not typically defined as an attitude. 2) In the discussion the Authors might add a comment on the influence of current withdrawal of compulsory vaccination of medical stuff in Italy.
Another interesting practical point for discussion is possible: carrying out a follow up of the study in the current epidemiological and legal situation.
Author Response
Dear reviewer,
Thank you for your valuable advice. After reading your comments we edited our work in the following points. A detailed answer to all the comments and the changes made are reported below:
Comment: “It would be helpful to provide a bit more detailed information regarding how attitudes were assessed (questionnaires?) as raw data from database regarding the fact who, when and where underwent vaccination is not typically defined as an attitude.”
Response: Thank you for your valuable comment, we did not assess attitudes, rather we evaluated the behaviour MRs have towards vaccination against COVID-19; the title of the manuscript has been changed to be more precise and more in line with the aim of this study. We changed attitudes with behaviours in the manuscript accordingly. Thank you so much for pointing this out to us.
Comment: “In the discussion the Authors might add a comment on the influence of current withdrawal of compulsory vaccination of medical stuff in Italy.”
Response: Thank you, this is a very interesting point and we have added this to our discussion (lines 278-289).
Comment: “Another interesting practical point for discussion is possible: carrying out a follow up of the study in the current epidemiological and legal situation.”
Response: Thank you for this input, this is indeed a very interesting query, and one we intend to further investigate in future studies; we have added a sentence detailing this intent (lines 289-295).
Round 2
Reviewer 1 Report
All my comments have been professionally addressed.
Reviewer 2 Report
Dear authors, thank you for considering my comments and trying to clarify when possible.